# The evolutionary history of topological variations in the CPA/AT transporters

**Govindarajan Sudha**[1][◉], **Claudio Bassot**[1][◉], **John Lamb**[1], **Nanjiang Shu**[2], **Yan Huang**[3], **Arne Elofsson**[1]*

**1** Science for Life Laboratory, Department of Biochemistry and Biophysics, Stockholm University, Solna, Sweden, **2** Bioinformatics Short-term Support and Infrastructure (BILS), Science for Life Laboratory, Sweden, **3** Science for Life Laboratory, Karolinska Institutet, Stockholm University, Solna, Sweden

◉ These authors contributed equally to this work.
* arne@bioinfo.se

**Data Availability Statement:** The results from this work are available as a database name CPAfold (http://cpafold.bioinfo.se/). All scripts are available from the following GitHub repositories https://github.com/ElofssonLab/TMplot, and https://

## Abstract

CPA/AT transporters are made up of scaffold and a core domain. The core domain contains two non-canonical helices (broken or reentrant) that mediate the transport of ions, amino acids or other charged compounds. During evolution, these transporters have undergone substantial changes in structure, topology and function. To shed light on these structural transitions, we create models for all families using an integrated topology annotation method. We find that the CPA/AT transporters can be classified into four fold-types based on their structure; (1) the CPA-broken fold-type, (2) the CPA-reentrant fold-type, (3) the BART fold-type, and (4) a previously not described fold-type, the Reentrant-Helix-Reentrant fold-type. Several topological transitions are identified, including the transition between a broken and reentrant helix, one transition between a loop and a reentrant helix, complete changes of orientation, and changes in the number of scaffold helices. These transitions are mainly caused by gene duplication and shuffling events. Structural models, topology information and other details are presented in a searchable database, CPAfold (cpafold.bioinfo.se).

## Author summary

The availability of experimentally solved transmembrane transport structures are sparse, and modelling is challenging as the families contain non-canonical transmembrane helices. Here, we present structural models for all families of CPA/AT transporters. These proteins are then classified into four "fold-types", including one novel fold-type, the reentrant-helix-reentrant fold type. We find extensive structural variations within the fold with members having from three to fourteen transmembrane helices. We explore the evolutionary mechanisms that have shaped the topological variations providing a deeper understanding of membrane protein structure and evolution. We also believe our work could serve as a model system to understand the evolution of topology variations for other membrane proteins.

github.com/gsudha/CPA_AT_database/. All data is available from https://doi.org/10.6084/m9.figshare. 14575626.v1. A user friendly interface to the data is available from https://cpafold.bioinfo.se/.

**Funding:** We thank the Swedish National Research Council (https://vr.se) for a grant (#2016-03798) to AE, and a grant from Knut and Alice Wallenberg Foundation (https://kaw.wallenberg.org/) for financial support. Salaries for AE, NS, SG, CB and JH were provided by Stockholm University, either through faculty funding or through the grants mentioned above. The funders had no role in study design, data collection and analysis, decision to publish, or preparation of the manuscript.

**Competing interests:** The authors have declared that no competing interests exist.

# Introduction

Proteins belonging to the Pfam CPA/AT clan (monovalent cation-proton antiporter/anion transporters) transport: ions, amino acids, and other charged compounds [1–4]. Due to their functional importance, these transporters are ubiquitously present in all three kingdoms of life [3, 5–7]. In humans, these transporters are associated with pathological conditions such as intestinal bile acid malabsorption, ischemic and reperfusion injury, heart failure and cancer [8, 9]. Therefore, these transporters serve as important drug targets. [10, 11].

In the Transporter Classification Database (TCDB), these transporters are classified into the CPA- and the BART-superfamily. In contrast, Pfam[3] and OPM[3, 12] group all families into a single superfamily ("CPA/AT clan" and "Monovalent cation-proton antiporter", respectively). PDB structures are available only for five families in the Pfam CPA/AT clan, with topologies starting from 10 transmembrane helices (TM) in the SBF family [13] to 12 TM in $Na^+/H^+$ Antiporter 1 and OAD beta [14, 15], and 13 TM of $Na^+/H^+$ Exchanger and 2HCT [6, 16].

All known structures of CPA/AT transporters consist of two inverted symmetric ***repeat units*** Fig 1A and 1B, which are essential to enable the different conformational states necessary for the transport mechanism [17–20]. Each of the repeat units can be further divided into two structurally distinct parts, the ***scaffold*** and the ***core subdomain*** (Fig 1A and 1B *[6]*). Two scaffold and core subdomains come together in structure to give rise to the complete scaffold and core domains Fig 1A and 1D.

The scaffold domain is involved in dimerization and also interacts with the core domain. The interface between the two domains forms an aqueous cavity where the substrate binds. The core domains generally consist of six transmembrane helices (three from each subdomain), with the middle helix of each subdomain being a non-canonical helix, either in the form of a broken or a reentrant helix. The broken helix is a transmembrane helix that crosses the membrane, but contains a discontinuity in the alpha-helix, forming a loop within the membrane Fig 1A. In contrast, the reentrant helix does not cross the membrane and, therefore, enters and exits from the same side Fig 1B. Two non-canonical helices of the same type (Broken or reentrant), from the N- and C-terminal core subdomain respectively, interact to form a polar non-helical part in the center of the membrane. This polar region is capable of binding and transporting ions [21] and transfer them to the other side of the membrane using an elevator mechanism [22].

To investigate the structural features of the protein families, we perform homology and ab initio modelling. Homology modelling was performed using Modeller [23] from the HHpred webserver [24] and ab initio-modelling with trRosetta [25]. Although homology modelling is usually the first choice for protein modelling, ab initio models can nowadays also be of high accuracy [26]. Here, we find that trRosetta [25, 27] can be used to model these transporter families where homology modelling fails. Most importantly, we identify a novel fold, named the reentrant-helix-reentrant fold, present in three families and describe their evolutionary history. The novel fold is characterized by the fusion of two reentrant helices in a single region wherein the homologous resolved structures of the reentrant helices are separated in two symmetric domains. To facilitate for other researchers, we present all models and alignments in a searchable database, CPAfold (http://cpafold.bioinfo.se).

# Results and discussion

## Topology annotation and protein structural modelling led to the identification of four fold-types in the CPA/AT transporters

A few evolutionary related Pfam families were added to CPA/AT clan to form the CPA/AT transporter dataset, for details see the material and methods section, this brings the final

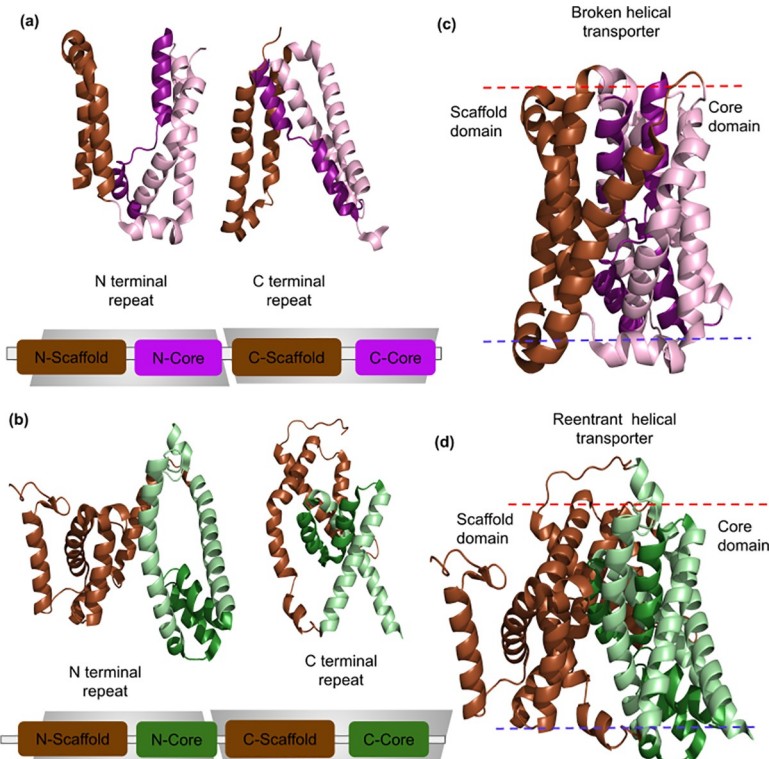

**Fig 1. General structure of CPA/AT transporters.** (a) Example of broken helical transporter (Sodium bile acid symporter: PDB 4n7w) shown as N- and C-terminal repeat. Each repeat unit is composed of a scaffold and a core subdomain. (b) Example of reentrant helical transporter (Sodium citrate symporter: PDB 5a1s) shown as N- and C-terminal repeat. (c) In structure space, both the N- and C-terminal subdomains form the scaffold and core domains, respectively. The lipid bilayer is colored red and blue to denote outside and inside. The scaffold and core domain are colored brown and purple respectively. (d) The scaffold and core domain are colored brown and green respectively. The broken and reentrant helices are shown in darker shades compared to other helices of the core domain for easy visualization.

dataset to include 23 protein families Table 1. Next, using the integrated pipeline we predict the topologies (including broken and reentrant helices) of all families. For topology annotations, we use a combination of evolution guided topology prediction, similarity to families with known structure, the difference in positively charged residues present in inside and outside loops in different topological models, and structural modelling. This effort allows us to determine the topology of the proteins in all Pfam families, resulting in a wide range of topologies from 3 to 14 helices. Table 1 Topologies identified in this work are in agreement with earlier experimental topology annotations [28–30].

The quality of the trRosetta models was estimated with Pcons Table 1. A benchmark for the Pcons scores of the models having known structure was performed in S1 Fig. In S1 Table are reported the Meff score (number of sequences clustered at 62% identity) of the MSA used for the model generation and the percentage of the satisfied contacts: both the scores correlate directly with the quality of the trRosetta models.

On the basis of the models, we grouped CPA/AT transporter families belonging to the same fold-type based on their topology and structure. Here, we define "fold-types" to be structurally similar repeat units found in a group of families. We find four different fold-types Table 1. Families that belong to the BART superfamily from TCDB belong to the BART fold type, while families from the CPA superfamily are classified into three fold-types: CPA-broken,

**Table 1. Annotation of topology and subdomains for families in the CPA/AT transporters.** The orientation ($N_{in}$, $N_{out}$) describes the location of the N-termini of the protein. Comparison of classification of families from this work, TCDB and Pfam are shown. The name in bold preceding the semicolon denotes superfamily, the name following the semicolon denotes the family. Families with unassigned superfamily start with a Semicolon. H, RH and BH indicate transmembrane helix, Reentrant helix and Broken helix respectively. When a homologous template was found with an E-value better than $1^*10^{-3}$ it was shown. The Pcons scores for the trRosetta models are also listed.

| No | Families belonging to CPA/AT transporters | Pfam | TCDB | Topology | Structure (PDB ID) | Template | PDB template E-values | trRosetta models Pcons score |
|---|---|---|---|---|---|---|---|---|
| | | | BART fold-type | | | | | |
| 1 | SBF_1 | **CPA/AT;** SBF (PF01758) | **BART superfamily;** Bile Acid: Na$^+$ Symporter -BASS Family, Arsenical Resistance-3 (ACR3) | 8H-2BH-$N_{in}$ | 4n7w | | | |
| 2 | SBF_2 | | | 7H-2BH-$N_{in}$ | | 4n7w_A | $1.1^*10^{-39}$ | 0.79 |
| 3 | SBFlike | **CPA/AT;** SBF_like (PF13593) | **BART superfamily;** Bile Acid: Na$^+$ Symporter -BASS | 8H-2BH-$N_{in}$ | | 4n7w_A | $4.7^*10^{-4}$ | 0.81 |
| 4 | KdgT | **CPA/AT;** KdgT (PF03812) | **BART superfamily;** 2-Keto-3-Deoxygluconate Transporter (KdgT) | 8H-2BH-$N_{in}$ | | 3zux_A | $2.6^*10^{-15}$ | 0.87 |
| 5 | Mem_trans | **CPA/AT;** Mem_trans (PF03547) | **BART superfamily;** Auxin Efflux Carrier (AEC) | 8H-2BH-$N_{out}$ | | | | 0.69 |
| 6 | Sbt_1 | **CPA/AT;** Sbt_1 (PF05982) | ;The Na+-dependent Bicarbonate Transporter (SBT) Family | 8H-2BH-$N_{out}$ | | | | |
| | | | CPA-broken fold-type | | | | | |
| 7 | Na_H_antiport_1 | **CPA/AT;** Na_H_antiport_1 (PF06965) | ; NhaA Na$^+$:H$^+$Antiporter (NhaA) | 10H-2BH-$N_{in}$ | 1zcd | | | |
| 8 | NA_H_Exchanger_1 | **CPA/AT;** Na_H_Exchanger (PF00999) | **CPA Superfamily;** Monovalent Cation:Proton Antiporter-1 (CPA1) | 11H-2BH-$N_{ou}$ | 4bwz | | | |
| 9 | NA_H_Exchanger_2 | | **CPA Superfamily;** Monovalent Cation:Proton Antiporter-1 (CPA2) | 12H-2BH-$N_{in}$ | | 4czb_B | $2.4^*10^{-48}$ | 0.72 |
| | | | CPA-reentrant fold-type | | | | | |
| 10 | Asp_Al_Ex | **CPA/AT;** Asp_Al_Exchanger (PF06826) | **CPA superfamily;** Aspartate: Alanine Exchanger (AAEx) | 10H-2RH-$N_{out}$ | | 5a1s_B | $3.3^*10^{-13}$ | 0.68 |
| 11 | Glt_symporter | **CPA/AT;** Glt_symporter (PF03616) | **CPA superfamily;** Glutamate: Na$^+$ Symporter (ESS) | 10H-2RH-$N_{out}$ | | 5a1s_B | $8.5^*10^{-38}$ | 0.94 |
| 12 | DUF819 | **CPA/AT;** DUF819 (PF05684) | | 10H-2RH-$N_{ou}$ | | 5a1s_B | $1.8^*10^{-42}$ | 0.93 |
| 13 | AbrB | **Membrane_trans;** AbrB (PF05145) | | 10H-2RH-$N_{in}$ | | 5a1s_B | $1.80^*10^{-8}$ | 0.93 |
| 14 | 2HCT | ; 2HCT (PF03390) | ; 2-Hydroxycarboxylate Transporter (2-HCT) family | 11H-2RH-$N_{in}$ | 5a1s | | | |
| 15 | OAD_beta | **CPA/AT;** OAD_beta (PF03977) | **CPA superfamily;** The Na+-transporting Carboxylic Acid Decarboxylase (NaT-DC) Family | 10H-2RH-$N_{out}$ | 6iww | | | |
| | | | Reentrant-helix-reentrant fold-type | | | | | |
| 16 | PSE_1 | **CPA/AT;** Cons_Hypoth698 (PF03601) | **CPA superfamily;** Putative Sulfate Exporter (PSE) | 9H-2RH-$N_{in}$ | | 5a1s_B | $1.3^*10^{-19}$ | 0.96 |
| 17 | PSE_2 | | | 11H-2RH-$N_{in}$ | | 5a1s_B | $3.1^*10^{-13}$ | 0.73 |

(*Continued*)

**Table 1.** (Continued)

| No | Families belonging to CPA/AT transporters | Pfam | TCDB | Topology | Structure (PDB ID) | Template | PDB template E-values | trRosetta models Pcons score |
|----|----|----|----|----|----|----|----|----|
| 18 | LysAB | **CPA/AT;** Lys_export (PF03956) | ;The Lysine Exporter (LysO) Family | 8H-2RH-$N_{out}$ | | | | 0.52 |
| 19 | LysA | | | 3H-$N_{out}$ | | | | 0.96 |
| 20 | LysB | | | 5H-2RH-Nin | | | | 0.94 |
| 21 | LrgAB | ;LrgB (PF04172) | | 11H-2RH-$N_{in}$ | | | | 0.52 |
| 22 | LrgB | ;LrgB (PF04172) | ; The LrgB/CidB Holin-like Glycolate/Glycerate Transporter (LrgB/CidB/GGT) Family | 6H-2RH-$N_{out}$ | | | | 0.94 |
| 23 | LrgA | ;LrgA (PF03788) | ;The CidA/LrgA Holin (CidA/LrgA Holin) Family | 4H-$N_{in}$ | | | | 0.86 |

CPA-reentrant or Reentrant-helix-reentrant (RHR). Crystal structures are available for at least one family in three out of four fold-types but not for the RHR fold-type. The identification and the evolution of this new fold type are discussed in detail in the following section.

All the data generated from this work is made available as "CPAfold database" (http://cpafold.bioinfo.se). The CPAfold database can be useful to experimentalists interested in studying the structure, function, mutation and design of any protein belonging to this fold. 3D structure models are available for all the families in this fold. Functional and mutational studies can be supported by these models. Reentrant motifs for the reentrant type transporters have been identified. Full length and repeat level topology and sequence alignments of a particular protein family with all the other families in the fold are provided. This becomes helpful for the comparison of the structure and function of related families of interest.

Users can check which family their protein of interest belongs to, as well as other details such as representative sequence, topology, multiple sequence alignments, Core and scaffold domain annotation, KR-bias, fold-type classifications are also available.

## The Reentrant-helix-reentrant fold-type

Through ab initio protein modelling, we identified the novel reentrant-helix-reentrant (RHR) fold. We found three families with this fold type; Putative Sulfate Exporter (PSE), LrgAB operon proteins and the Lys_export family.

## Putative Sulfate Exporter (PSE)

The PSE family (Pfam: PF03601, Cons_hypoth698) exists in two topological forms with 9 and 11 transmembrane domains (TM), here referred to as PSE_1 and PSE_2. Their predicted structures are almost identical except for the two additional N-terminal TM helices in PSE_2. The PSE family share significant sequence similarity with the 2HCT family, E-value $1.3^{*}10^{-19}$, using HHsearch [31], Table 1. There is one known structure (PDB: 5a1s) in the 2HCT family. However, the two trRosetta models for the PSE family show substantial differences compared to the corresponding homology models based on the 5a1s template Fig 2A and 2B. In particular, the homology models lack the second reentrant helix, leaving a central cavity in the model. This means that in the homology model the active site is not complete. In contrast, the trRosetta model of PSE_1 contains two reentrant helices, and a potential active site can be identified.

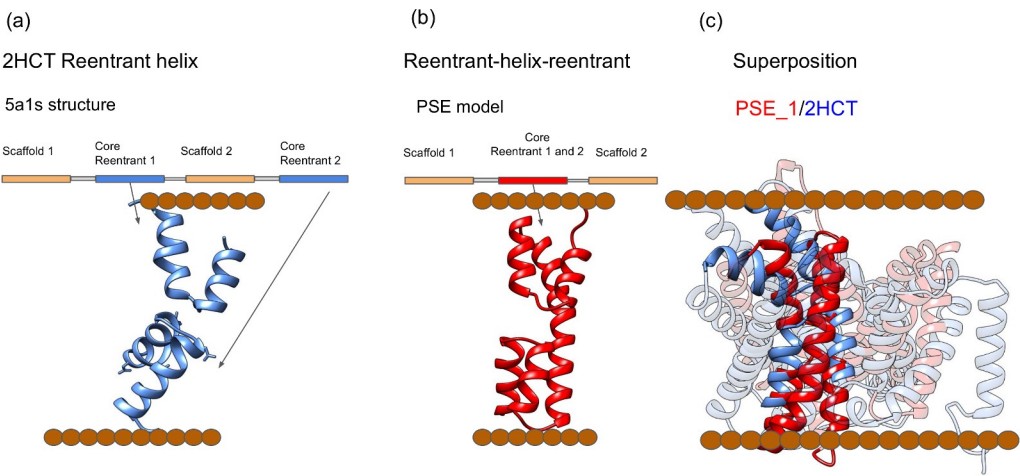

**Fig 2. Comparison between PSE_1 (Reentrant-helix-reentrant fold-type) and 2HCT (CPA-reentrant fold-type).** The brown dots represent the localization of the membrane a) An example of an internal symmetry with two reentrant helices (2HCT:5a1s). b) PSE_1 model. c) Superposition between the models of PSE_1 (red) and the 2HCT structure (blue).

Three additional observations confirm the reliability of the PSE_1 trRosetta model: (1) alternative contact predictions methods provide very similar models S2 Fig, (2) a model of the E. coli member of PSE (YieH) has been presented earlier with an identical fold [32], (3) the recently added protein model from Pfam predicts a similar fold (see Pfam website).

Structurally, the core of the trRosetta PSE_1 and 5a1s from the 2HCT family are aligned, see Fig 2C. However, in PSE_1, only a single transmembrane helix separates the two reentrant helices, while in 2HCT, five helices are separating the two reentrant helices. Thus, the fold of the C-terminal subdomain is different. This reentrant-helix-reentrant fold is also distinct from all known CPA/AT structures and PSE_1 is to the best of our knowledge the first example of CPA/AT transporters without an internal symmetry.

## LrgAB operon proteins and the Lys_export family

The LrgAB operon proteins and the Lys_export family also contain an RHR fold. However, these families exist as both monomers and dimers. In Prokaryotes, LrgB contains the RHR fold but interacts with the auxiliary smaller subunit LrgA (Pfam: PF03788) which contains four transmembrane helices. While in eukaryotes, it exists as a fused monomeric form LrgAB (Pfam: PF04172). Similarly, the Lysine exporter (Pfam: PF03956) has two topological forms: a monomeric fused form LysAB and two shorter proteins LysA and LysB forming a dimer that interact with each other. LysB and LysAB contain the RHR fold while LysA is a short protein with three transmembrane helices. For more information about the LrgAB operon and Lys_export, please refer to the CPAfold database.

## Evolutionary relationships between the fold-types in the CPA/AT transporters

Sensitive MSA-MSA alignments of repeat units or full-length proteins between a pair of families reveal evolutionary relationships between the different fold-types. Each family MSA was searched (full length or repeat unit) using HHsearch against the MSA of all Pfam families in Pfam-A_v32. The output is an alignment of the query family MSA and the Pfam family MSA, these query-template MSA are referred to as MSA-MSA alignments.

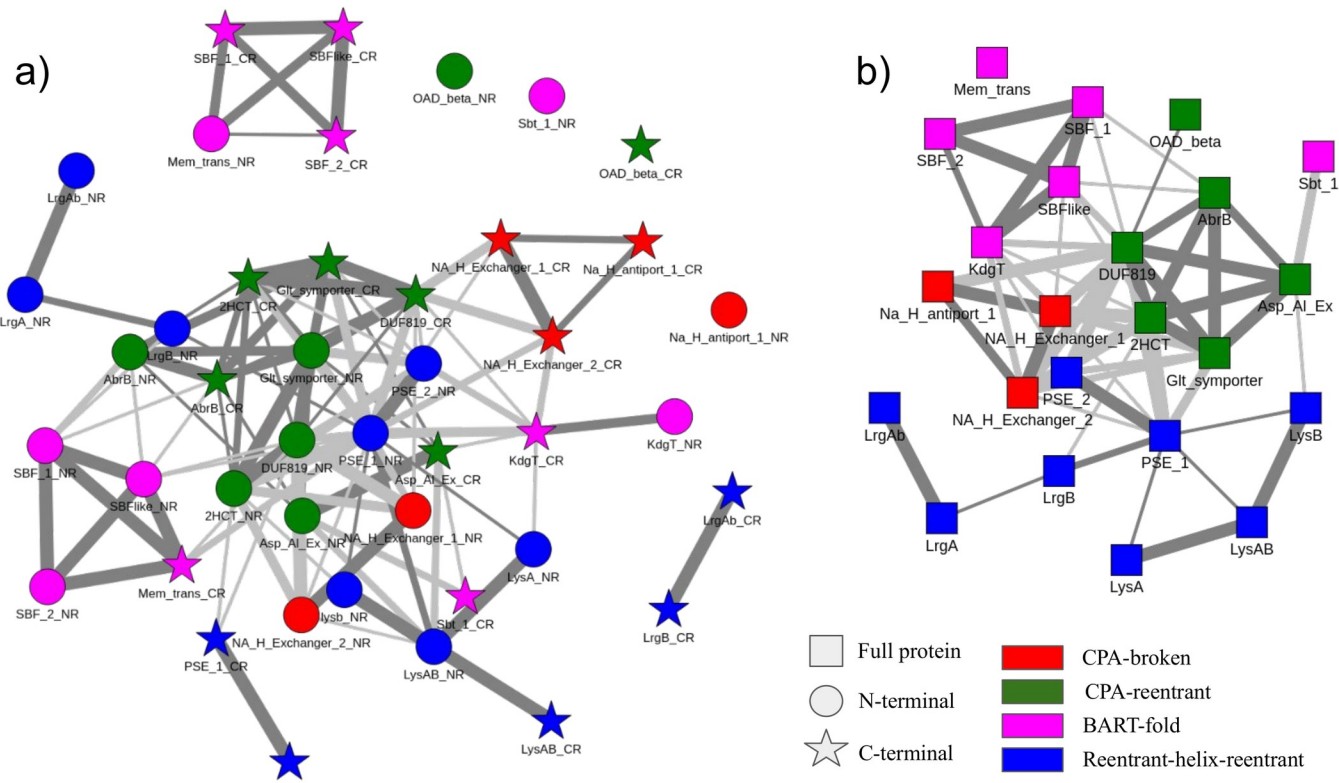

**Fig 3.** Visualization of the relationship between the families by a network linking families with significant similarities in the MSA-MSA alignments (E-value < 0.1 thin edges, E-value <0.001 thick edges) for repeat units (right) and full-length families (left). Each family/repeat is colored according to the fold type, Magenta: BART-fold, Green: CPA-reentrant, Red: CPA-broken and Blue: reentrant-helix-reentrant fold type. In the repeat unit network, a) N-terminal units are circular and C-terminal units are star-shaped. In the full protein network. b) The full proteins are square.

Scores of the alignments are used to generate a weighted network. Both the full length and repeat level networks Fig 3A and 3B show one major cluster consisting of the CPA-reentrant, CPA-broken families as well as the PSE family from the reentrant-helix-reentrant fold type. The families in BART fold-type and the other families in RHR fold-type are only loosely connected to this main cluster. This clearly shows that CPA-broken and CPA-reentrant fold-types are evolutionarily closest to each other. The evolution of the novel RHR fold-type from CPA-reentrant is also evident, at least for PSE. The exact relationship of the outlier families is however not completely clear from the evolutionary analysis.

The repeat network also clearly highlights the shuffling of the domains in the mem_trans family compared to the other BART families, discussed below. The repeat level network also helped to infer the various evolutionary mechanisms responsible for the evolution of different fold-types.

Evolution can occur both at the repeat level and involving the entire protein. For example, it is clear from the network discussed previously that evolution within the fold-types typically occurs at full length, as two N-terminal repeat units in different families are closer to each other than they are to the C-terminals.

## Different types of topology transitions in CPA/AT transporters

**1) Loop-reentrant transition due to truncation of the repeat unit.** The networks in Fig 3 clearly show the evolutionary relationship between the reentrant-helix-reentrant fold-type,

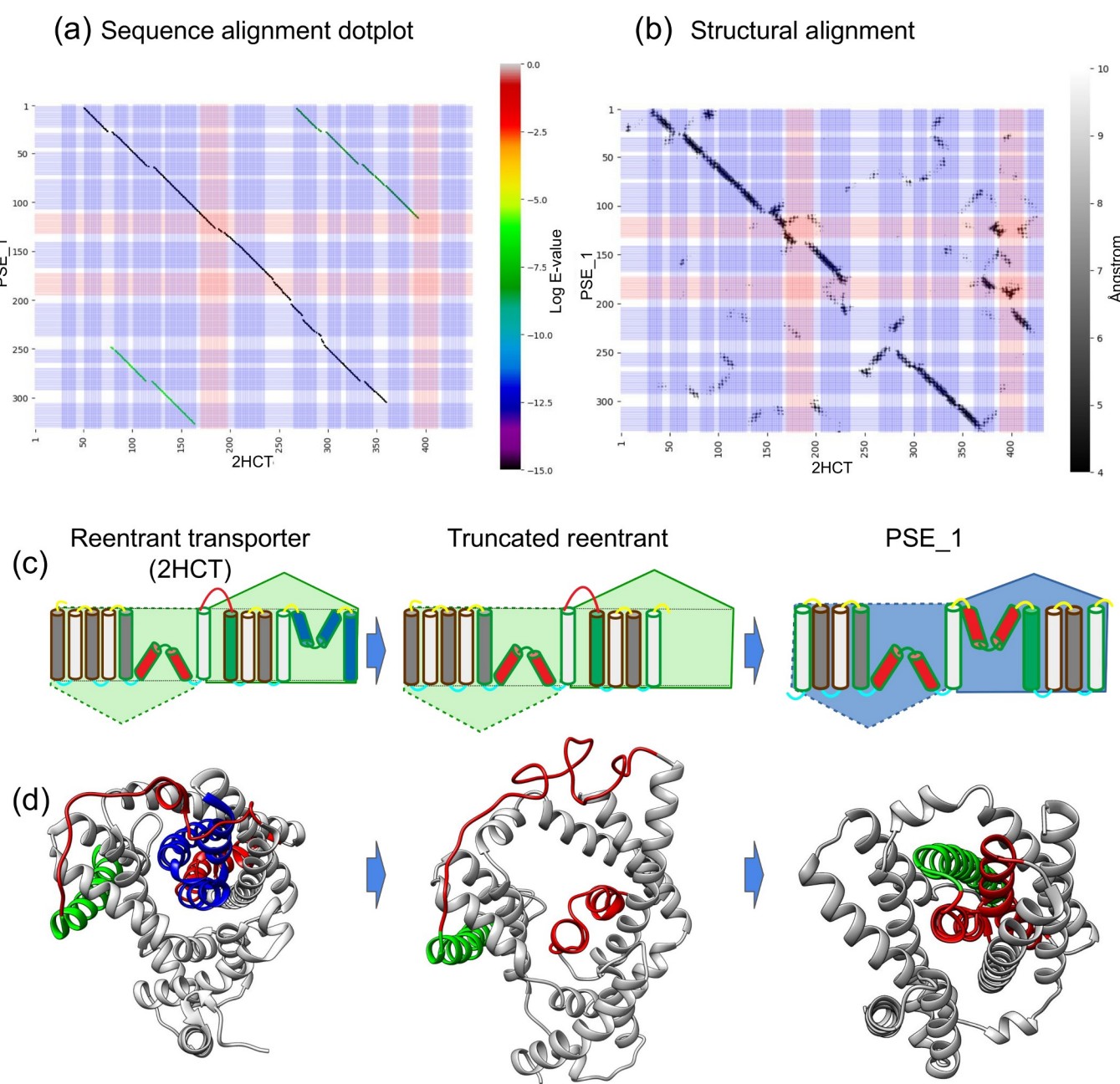

**Fig 4. Transition and alignments between 2HCT and PSE_1.** a) A "Dotplot" between PSE_1 and 2HCT: The blue lines represent the TM helices and the red one the reentrant helices, the color spectra show the Log of E-value of the alternative alignments. b) The panel shows an all-vs-all distance matrix between the trRosetta models and the 2HCT PDB structure 5a1s. The greyscale indicates the distance between the mutual distances between the protein residues from 4 to 10 Å. b) Schematic representation of the topology of the protein with its corresponding structure below. The TM helices (in-out) and (out-in) are colored white and grey respectively. The reentrant helices after the transition are colored red. The helix 7 that changes position after the transition is colored green. The part of 2HCT putatively lost in the truncation is colored blue. c) Corresponding protein structures.

PSE_1 family and the CPA-reentrant fold-type family, Citrate Symporter family (2HCT). From the alignment between PSE_1 and 2HCT (identity 14.5%, 316 aligned positions), it can be speculated that PSE is a truncated version of *2HCT*, Fig 4A and 4B. The sequence alignment shows that PSE_1 aligns from TM2 to TM10 of *2HCT*, i.e., both scaffold domains as well as the N-terminal core domain are conserved (except TM1) but only the first TM helix of the second

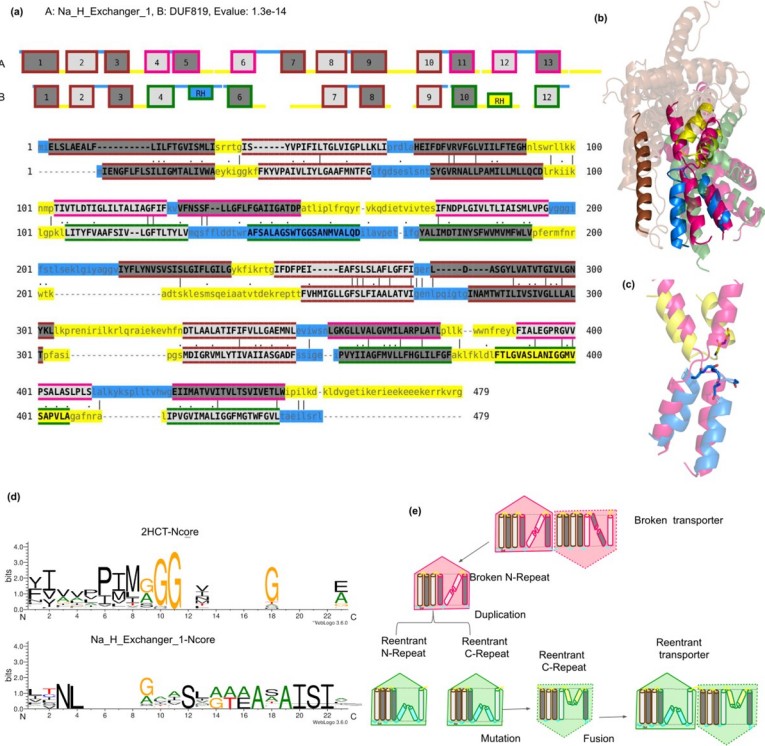

**Fig 5. Broken-Reentrant transporter transitions by duplication of repeat unit.** (a) Sequence and topology alignment between Na_H_Exchanger_1 (CPA-broken fold-type) and DUF819 (CPA-reentrant fold-type). All the transmembrane helices are numbered sequentially and the reentrant helix is denoted as RH (b) Structure superposition of broken and reentrant transporter. The extra scaffold helix and the broken-reentrant transition are highlighted in bright colors. (c) A zoomed-in figure of broken (pink) and reentrant helix (yellow and blue) is shown. The glycines are shown in stick representation. (d) The aligned positions of the broken and reentrant N-core helix are represented as sequence motifs (e) Cartoon representation showing the events of duplication and mutation in reentrant transporters leading to the transition of broken-reentrant transporters.

core domain is present in PSE. The alignment of TM1-5 of PSE to TM7-11 of 2HCT reveals an internal symmetry; however, the alignment contains a gap, covering the second reentrant helix and TM6 (residues 169–248) of PSE_1. This region is instead structurally aligned to the second reentrant helix and TM11 of 2HCT.

We suggest that the novel fold could have arisen through a rescue mechanism of a truncated reentrant protein from the 2HCT family. The truncated transporter would not be functional as the active site is not complete. However, the structure could have been resurrected by the adaptation of a novel fold in which; (i) TM7 (green in Fig 4C and 4D) moved into the center of the protein and (ii) the connecting loop (red in Fig 4C and 4D) became a novel reentrant helix. This ancestral protein would then have the topology of PSE. The LrgAB operon and Lys exporter families could then evolve from this ancestral protein by terminal duplications and rearrangements, see Fig 5. However, the sequence identity is so low between these families and all other families, so we cannot exclude that two families have evolved independently from some other ancestral reentrant protein.

An interesting question is if the ancestral protein was functional. If it was truncated and no reentrant region was present it could certainly not function as a transporter. However, we do believe that it is not necessary to assume that the ancestral protein was non-functional. It is possible that the new reentrant region in PSE might exist as a minor state even in a full-length ancestral protein.

The reentrant-helix-reentrant fold has not been described before in the CPA/AT Pfam clan, but the fold is present in a protein in the SLC1/EAAT transporter family [33, 34] as well as in the predicted structure of human protein Tmem41b [35]. If these families have a common origin or are the results of convergent evolution is unfortunately not possible to deduce as we cannot detect any sequence similarity between these families.

**2) Broken-reentrant transition by repeat duplication.** DUF819 (CPA-reentrant fold-type) and Na_H_Exchanger_1 (CPA-broken fold-type) are the two closest related families Fig 5A.The N-terminal repeat of Na_H_Exchanger family (CPA-broken fold-type) is closely related to both the N- and C-terminal repeat units from DUF819 (CPA-Reentrant fold-type), see S3A and S3B Fig. Broken -reentrant transition involves many changes. First, a gain or loss of a helix at the C-terminal scaffold subdomain is observed in order to maintain the inverted nature of the repeat units Fig 5A and 5B.

The CPA-reentrant is enriched in glycines and sometimes prolines in the core motif with low hydrophobicity thereby aiding the formation of the loop Figs 5C, 5D and S4A.The core sequence motifs for all the families are available in the CPAfold database.

It is well known that the cytoplasmic sides of membrane proteins are enriched in positive residues (K and R) [36]. Thirdly, the transition between a reentrant and a broken helix changes the orientation of the last helix of the core subdomain, thereby also changing the orientation and packing of all the following helices S4B and S4C Fig. Therefore, the number of positively charged residues in the surrounding loops should change upon transition.

The structural basis of topological transitions discussed previously holds true for all the other pairs of families belonging to the CPA-broken and CPA-reentrant fold-types. Our sequence and structural analysis show that the initial transition between the broken and reentrant transporters occurred by duplication of the repeat. Therefore, we propose that the broken-reentrant transition includes these following steps: (i) Mutations in the N-terminal repeat of the CPA-broken repeats leading to reentrant helix (ii) Duplication of the repeat unit, (iii) fusion of the repeats, and (iv) mutations to change the orientational preference of one of the repeats to form a functional reentrant transporter Fig 5E.

**3) Changes in orientation in the CPA-reentrant-fold by internal duplication.** Families from the same fold-type can have opposite orientations Figs 6A and S5. All CPA/AT transporters have an internal symmetry, i.e., their evolutionary history starts from an internal duplication of repeat units at some point of their evolutionary history. However, in most families, the internal duplication is more ancient than the divergence between protein families, as the N-terminal repeats from different families are more similar than repeats within the same protein family (i.e., between N-terminal repeat and C-terminal repeat). Exceptions to this rule are two CPA-reentrant fold-type families, Asp-Al_Exchanger and AbrB Fig 6B. These families have a very high similarity between the N- and C-terminal repeat units, a clear signal of a recent internal duplication. The reentrant core motif is enriched with both glycines and prolines Fig 6C.

MSA-MSA alignments and repeat network shows the C-terminal repeat of Glt_symporter is evolutionarily close to both the repeat units of AbrB. Therefore, a scenario describing the evolutionary change starts from the C-terminal repeat of an ancestral Glt-symporter S6 Fig. This repeat unit is then internally duplicated and fused followed by mutations to change the orientation of the C-terminal-repeat, see Fig 6D. Asp_Al_exchanger also has a recent internal duplication, most likely caused by an internal duplication starting from the N-terminal repeat unit.

**4) Changes in orientation in BART-fold by shuffling of repeats.** The Mem_trans family has an opposite orientation compared to all the other families of the BART fold-type (e.g., SBF_like) Table 1. The N-terminal repeat of the SBF_like family is similar to the C-terminal-repeat of Mem_trans and vice versa Figs 7 and S7. This is also clear from the repeat network

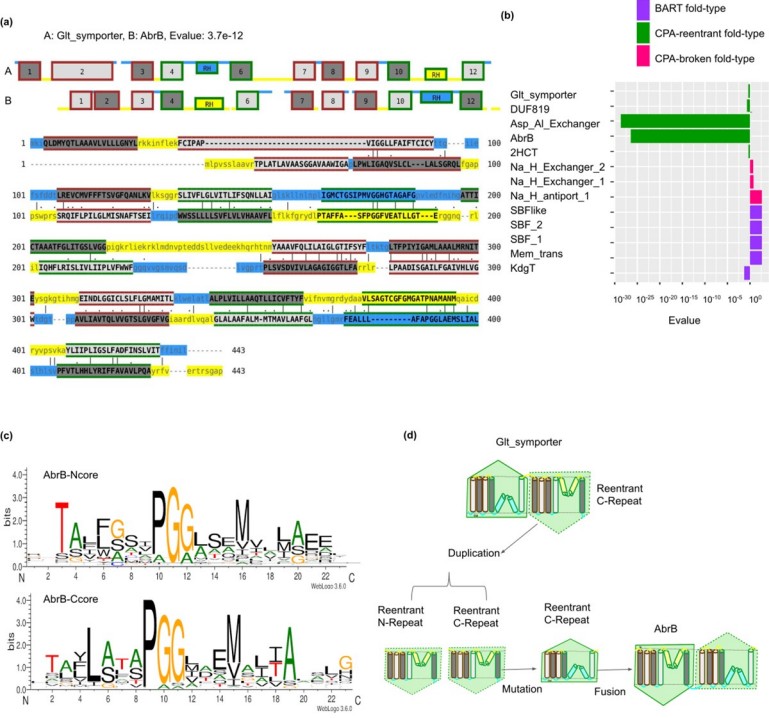

**Fig 6. Change in orientation in reentrant transporters by internal duplications.** (a) Sequence and topology alignment between transporters from CPA-reentrant fold-type showing the change in orientation. All the transmembrane helices are numbered sequentially and the reentrant helix is denoted as RH. (b) Sequence similarity between N- and C-terminal repeats represented by E-values in different families with fold-types containing symmetric repeat units are shown (c) Reentrant N- and C-terminal core helix motif (d) Cartoon representation showing the events of duplication from C-terminal repeat of reentrant transporter and subsequent internal duplication leading to change in orientation.

and the pairwise repeat alignments. This reciprocal similarity between the repeat units shows that there has occurred a recent shuffling of the repeat units causing the change in orientation, see Fig 7C.

**5) Gain/loss of scaffold helices.** It is generally assumed that the topology is conserved within a family [37]. However, members of SBF, Na_H_Exchanger and PSE families have two distinct topologies S8 Fig. Therefore, these families were split into two subfamilies based on their topologies Table 1. Topology variations between subfamilies/families within the same fold-type always show gain/loss of helices in the N-terminal scaffold subdomain S8 and S9 Figs. In contrast, topology variations in families between fold-types always have changes in their C-terminal scaffold domain. It may or may not have changed in the N-terminal repeat S10 Fig.

## Conclusion

In this work, we mapped and analyzed the vast variability in folds and transmembrane topologies in CPA/AT transporters. Fig 8 summarizes the variation and the evolution of the CPA/AT transporters. CPA/AT transporters generate a selective permeability across cell membranes and a key role in the substrate selection is played by the non-canonical-helices. Here we suggest that the transition between broken, reentrant and reentrant-helix-reentrant helices can be caused by multiple point mutations or a stop codon mutation. The new reentrant region might be present in an ancestral protein with multiple states or it might have appeared to

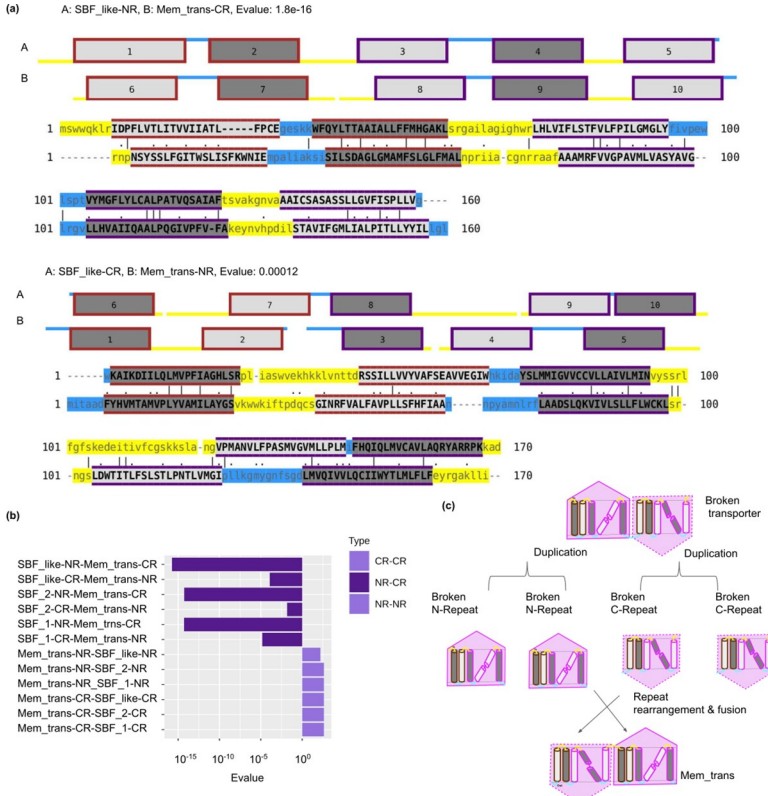

**Fig 7. Change in orientation in broken transporters by shuffling.** (a) Sequence and topology alignments, which shows the high similarity between N-terminal and C-terminal repeats of SBF_like and Mem_trans families and vice versa. All the transmembrane helices are numbered sequentially. (b) Sequence similarity, represented with E-values, between NR-NR, CR-CR and NR-CR repeats of two different families in the BART fold-type are shown. NR and CR refer to the N- and C-terminal repeats respectively. (c) Cartoon representation showing events of shuffling of repeats leading to change in orientation.

resurrect a non-functional protein. Transporters are also characterized by pseudo-symmetry in sequence and structure. We provided evidence of how the two repeats can evolve independently, shuffle and maintain transport functionality.

A final common feature of the transporters is their existence as dimer or trimer. The scaffold domain acts as an oligomerization interface and is the part of the proteins that show the highest variability in terms of topology. Among CPA/AT transporters indeed we identified variation in the topology even within the same protein family. Collectively, these pieces of evidence provide a complete picture of the surprising plasticity of membrane transporters.

# Materials and methods

## 1) Topology annotations of families and subfamilies

Our strategy to annotate topology and reclassify the Pfam CPA/AT clan into families/subfamilies involves the following five steps, also described in Fig 9. Identification of Pfam subfamilies, each with a unique topology and assignment of initial topologies.

i. Improved classification of CPA/AT transporters

ii. Generating a final topology.

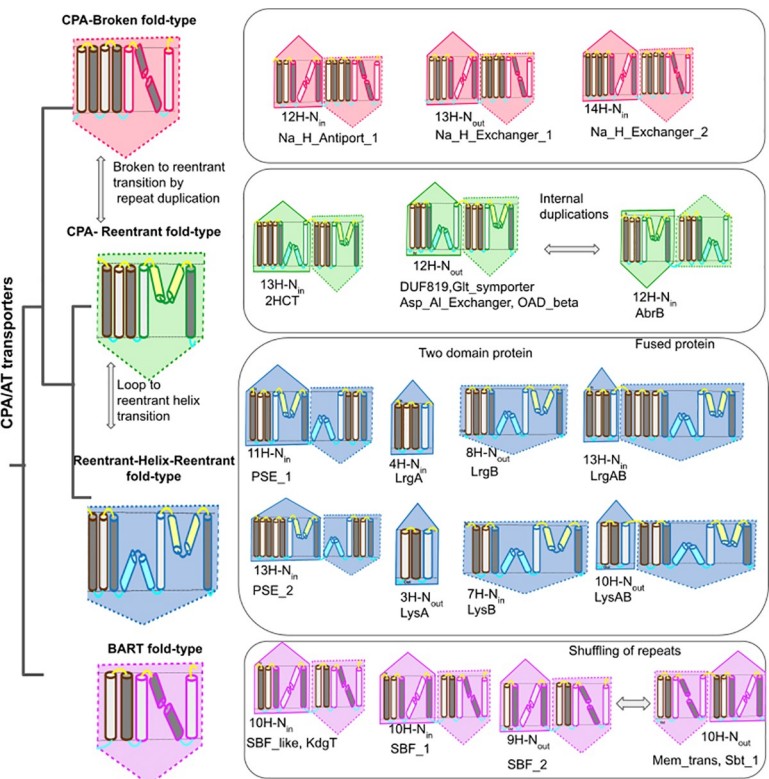

**Fig 8. Fold-types and their evolution in CPA/AT transporters.** The four fold-types of CPA/AT transporters are shown. Topologies, type of core helix (Broken/reentrant), the orientation of the N-terminal helix as well as other data are listed. All the evolutionary events responsible for the evolution of fold-types are summarized. The TM helices (in-out) and TM helices (out-in) are colored dark grey and white, respectively. Reentrant helices (in-in) and (out-out) are colored yellow and blue respectively.

iii. Identification of core, scaffold subdomains and repeat units from the known structure.

iv. Validation of Broken/reentrant type transporters by the positive inside rule.

**i) Identification of Pfam subfamilies, each with a unique topology and assignment of initial topologies.** We extracted reference proteome sequences from the 13 Pfam families in the CPA/AT clan [36, 38]. The reference proteomes are complete proteomes that represent taxonomic diversity. In particular, they include the well-studied model organisms that are of biomedical and biotechnological interest. Fragments, sequences with <75% Pfam domain coverage, and highly similar sequences (>90% identity) were excluded. The remaining sequences were clustered at 30% identity using blastclust [39] and aligned using Clustal Omega to generate a seed MSA[40]. Topologies for all the members of the families were predicted using TOPCONS2 [41]. It is important that seed MSA represents the full-length sequence to map the complete topology.

Predicted topologies were mapped on to the seed MSA to get a topology alignment. A phylogenetic tree is generated from the seed MSA using the FastTree program [42]. The seed MSA is reordered according to the branching order of the phylogenetic tree. This MSA is renamed as the "Reordered topology alignment". This procedure of reordering the topologies serves two purposes: (1) It helps to weed out the topology errors caused by topology prediction programs (2) Identify evolutionarily distinct clusters that show systematic topology variation in

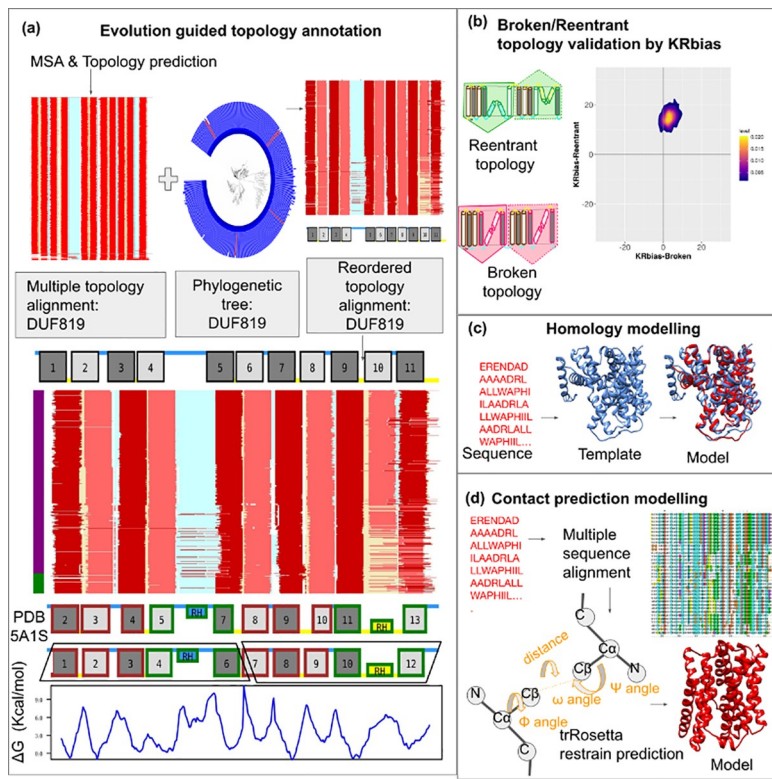

**Fig 9. An example showing the annotation of the topology of DUF819 family using our integrated pipeline.** The main steps involved in the pipeline are shown. (a) Evolution guided topology annotation: Topology predictions are mapped onto the seed MSA to obtain a multiple topology alignment. The phylogenetic tree from seed MSA is used to reorder the multiple topology alignment into a "Reordered topology alignment". This evolution guided topology prediction is used to infer initial topologies for a family. Generation of the final topology and annotation of core and scaffold subdomains is obtained by comparing to the known structure of sodium-citrate symporter (PDB id: 5A1S). The TM helices (in-out) and TM helices (out-in) are colored dark red/grey and light red/grey, respectively. Reentrant helices (in-in) and (out-out) are colored yellow and blue respectively. The inside and outside loops are colored yellow and blue respectively. The vertical bar is colored based on the taxonomy of the sequences (Bacteria: Purple, Archaea: Dark blue Eukaryotes: Green). Scaffold subdomains and reentrant core subdomains are colored brown and green, respectively. N- and C-terminal repeats are shown as black trapezoids. ΔG values describing the hydrophobicity [65] are obtained for the representative sequence and are plotted to the aligned residues in the representative sequence. (b) Validation of broken/reentrant transporters by using the KRbias for the DUF819 family. KRbias or positive inside rule is the enrichment of inside loops compared to the outside loops of a transmembrane protein[36, 46, 47]. The number of Lys (K) and Arg(R) amino acids in the inside and outside loops are compared. The expected correct topology would show a higher KR-bias in one of the two topology models (Broken/Reentrant). The KR-bias plot is shown as a 2D scatter plot. (c) Homology modelling of the representative sequence (d) Contact prediction modelling pipeline.

the predicted topologies. If the reordered topology alignment shows more than one topology, the MSA is then split based on the topology. The families are split into subfamilies that each have a unique topology. Despite combining the evolutionary information guided topology predictions, it was noted that the topology predictions sometimes consistently missed predicting either one or both broken/reentrant helices. Therefore, the topologies annotated using the reordered topology alignments were labelled as "Initial topologies". To assign correct topology to all families within the CPA/AT clan, we had to use additional steps Fig 9A.

**ii) Improved classification of the CPA/AT transporters.** A representative sequence for each family was selected. In the case of families with a known structure, this sequence was selected. In cases without a known structure, the top hit from a single search with HMMsearch [43] against Uniprot was used. The representative sequence of each family was searched

against Uniclust30 [38] using the HHblits program [44] with an E-value cut-off of 0.001 and 3 iterations generating the family MSA. The Meff score for the family MSA was calculated using cd-hit [45] with an identity cutoff of 0.62 and a word length of 4.

We used HHsearch version 3.2.0 [31] to find possible evolutionary relationships between the family MSA of the CPA/AT clan and other families in Pfam-A_v32.0 [3]. We wanted to search for new Pfam families that are not yet assigned to be part of CPA/AT clan.

Pfam clans are not always updated. We updated the Pfam clan CPA/AT by adding a few more Pfam families. HHsearch alignments clearly showed that LrgA (PF03788), LrgB (PF04172), AbrB (PF05145) and 2HCT (PF03390) are evolutionarily related to Pfam CPA/AT clan. 2HCT family has not been classified into any Pfam family. The AbrB family has been mis-classified into the Mem_trans clan in Pfam. New families that are added to the CPA/AT clan went through the previous step to annotate their initial topology and check if they have topology variations within the family.

**iii) Generating final topology.** The next step was to identify the missing, broken/reentrant helices if any. The representative sequence of the family/subfamily was searched against the PDBmmCIF70_22_May database using HHsearch [31] to compare the "initial topology" of the family with the topology derived from the crystal structure. PDBmmCIF70 is a HHsearch database that belongs to the HHsuite program. PDB_mmCIF70 are filtered with representative sequences with a maximum of 70% sequence identity selected. The query HMM is compared to the database of HMM-based on PDB chains to generate the query-template MSA-MSA alignments. Missing transmembrane helices (reentrant or broken) in the representative sequence were inferred from the alignment to the known structure.

Predicted topology and the topology from the crystal structure were mapped onto the pairwise MSA-MSA alignment to obtain a topology alignment. Transmembrane helices were considered to be aligned when at least five residues of both helices are aligned, as used before [46]. Otherwise, it is a, TM helix aligned to gap regions, TM helix aligned to inside/outside loops, TM helix aligned to signal peptide. The type is chosen based on the dominating composition in the segment of the TM helix that is aligned. Missing helices in the representative sequence were inferred when the TM helix in the topology with known structure was aligned to loops in the representative sequence. Since we have structural templates for both broken and reentrant type transporter, classification is based on the type of transporter with a known structure with the best hit (lowest E-value) S2 Table. Based on the classification of broken/reentrant type, missing helices were added, and the orientations were corrected. The final topology was then inferred for all families Fig 9A. All the helices, including the transmembrane helices and the non-canonical helices, are counted together to assign the topology. The orientation of the protein is assigned based on the N-termini.

**iv) Identification of core, scaffold subdomains and inverted repeat units from known structures.** Annotations of scaffold and core subdomains were taken from the literature of the Pfam families with known structure. Subdomain annotations were then transferred from the family with an available structure to the family with an unknown structure based on the definition of aligned TM helices described in the previous section Fig 9A.

**v) Validation of Broken/Reentrant type of transporters by the positive inside rule.** The "positive-inside rule" or KR-bias is the preferential occurrence of positively charged residues (lysine and arginine) at the cytoplasmic loops of transmembrane helices [36, 47, 48]. Therefore, it can be used to identify the orientation of the protein. A large-scale study with statistical observations of 107 organisms strongly supports the positive-inside rule. In general, KR bias can be applied to any genome for topology prediction [49, 50]. We counted the number of K (Lysine) and R (Arginine) starting from 10 residues inside the TM helix and up to 25 residues after the helix as this has been shown to contribute to the positive inside rule [51]. The KR-bias

is then calculated using the family MSA and comparing the number of KR in the inside and outside loops. Two models were made one representing the broken, and one the reentrant topology. This was then used to confirm the topology of all families/subfamilies. KR bias is calculated for all the helices in the full-length protein, and the expected correct topology would show a higher KR-bias Fig 9B.

### Generation of protein models and quality assessment

Topology annotations explained in the previous section were also supported and validated with homology or ab initio models for the representative sequence. The homology models were generated by Modeller [23] from an alignment and templates obtained from the webserver HHpred [24] Fig 9C, with the selected templates shown in Table 1.

The trRosetta models were generated running trRosetta locally [25]. Subsequently, we use the predicted distances and angles as input for pyRosetta [52]. We run pyRosetta twenty times obtaining twenty models from the same set of constraints Fig 9D. The quality of the contact predicted models were then evaluated using Pcons [53] giving as input the folder with the twenty models. The benchmark between the models with PDB structures and relative Pcons score is provided in S1 Fig.

In addition, for PSE_1 we used three alternative methods: RaptorX [54], DeepMetaPsicov [55] and PconsC4 [56]. In all these methods, we used the same MSA as input as for trRosetta. RaptorX was run from the web server while we used the pipeline we developed in Bassot et al. [57] that use the predicted contacts from PconsC4, DeepMetaPsicov and the Psipred 3.0 [58] secondary structure prediction as restraints for Confold [59]. All models have the same fold.

The topology annotation for all the families belonging to the Reentrant-helix-reentrant foldtype was carried out from the ab-initio models only, as we failed to annotate topology using evolution guided topology annotation because we could not recognize the second reentrant helix.

The Eukaryotic LrgAB protein was treated differently as this subfamily is present only in eukaryotes and is distinct from the prokaryotic members of the family. For the LrgAB protein, we generate the multiple sequence alignment using only eukaryotic sequences, generated from sequences retrieved from the Jackhammer web server [60]. The choice of the web server was made due to the ease of collecting the sequence of specific groups of organisms. Starting from the representative sequence of LrgAB (Swissprot: Q9FVQ4) we run three iterations with an E-value cutoff of $10^{-3}$ on the eukaryotic reference proteomes. The obtained full sequences were downloaded and used as a customized database for a local run of Jackhmmer [43] from where we obtained the family MSA.

### Generation of Dotplots

The multiple sequence alignments to generate the dotplot were created using HHsearch with three iterations and an E-value of $10^{-3}$ (Family MSA). The structural alignments were obtained with TMalign and the distances between the carbon α of the residues of the two proteins are calculated with Chimera [61]. Dotplots were generated using Matplotlib scripts available from the GitHub repository. In the sequence alignment dotplots, the colors correspond to the order of magnitude of the E-value of the alignment. The aligned dots are colored. The aligned residues were recovered from the HHsearch as described above. In the structural alignment, the shade of black corresponds to the distance between the aligned residues expressed in Ångstrom.

### Network analysis

The HHsearch program searches the query family MSA (Full length or repeat unit) against the MSA of all Pfam families in Pfam-A_v32. The output is an alignment of the query family MSA

and the Pfam family MSA. These query-template MSA are referred to as MSA-MSA alignments. The hit is considered to be N- or C-terminal repeat based on the alignment with query repeat. If bi-directional pairs of query-hit are obtained, the pair containing the lowest E-value is obtained. $\log_{10}$(E-values) were used to construct a weighted network using the python library NetworkX.

**3) Generation of topology alignments between families.**   MSA-MSA alignment between families obtained in the previous steps is converted into pairwise topology alignments. The topology and sequence alignment figure were generated. Some MSA-MSA alignments did not give rise to correctly aligned TM helices, due to uncertainty in introducing long gaps in full-length alignments. These cases are accompanied by correctly aligned structure alignments.

**4) Sequence motifs in the broken/reentrant helix.**   The middle helix (Broken/reentrant) of the N- and C-terminal core subdomain were extracted from the family MSA. Sequence motifs were generated using Weblogo program [62] to access the enriched amino acids in the broken and reentrant helices. Additionally, multiple sequence alignment consisting of the CPA-broken fold-type and CPA-reentrant fold-type was used to study the enrichment of amino acids and assessment of the type of mutations in both groups.

**5) Hydrophobicity (ΔG) and KR bias for broken and reentrant helices.**   The biological hydrophobicity scale (ΔG) and KR bias were calculated for all the proteins of families. Hydrophobicity of the broken/reentrant helix was calculated using DGpred [63, 64]. KR bias is calculated for the last helix of the N- and C-terminal core subdomains for all the proteins of families.

**6) Sequence similarity between repeat units.**   E-values between the N- and C-terminal repeats within the same family are obtained from the full-length alignments using the HHsearch program. These were carried out for all symmetry containing fold-types except the Reentrant-helix-reentrant fold-type.

**7) Structural superposition.**   Structure superpositions of the pairs of transporters were carried out separately for the core domain and scaffold domain. Structure superpositions were carried out using TMalign [65] and visualized using PyMol [66].

## Supporting information

**S1 Fig. Benchmark of the trRosetta models against the existing structure S2 Table of the CP/AT transporters.**
(PDF)

**S2 Fig. Models obtained by 4 different methods; trRosetta, RaptorX, DeepMetaPsicov and PconsC4.** a) The superposition among the four. b) trRosetta vs PcosnC4 c) trRosetta vs Deep-MetaPsicov d) trRosetta vs PconsC4
(PDF)

**S3 Fig. Sequence and topology alignments between Na_H_exchanger_1 and DUF819 families.** (a) Sequence and topology alignment between Na_H_Exchanger_1 N-terminal repeat and DUF819 C-terminal repeat (b) Sequence and topology alignment between Na_H_Exchanger_1 N-terminal repeat and DUF819 N-terminal repeat.
(PDF)

**S4 Fig. Molecular basis of Broken-reentrant helix transition.** (a) Density plot showing the hydrophobicity of broken and reentrant helices belonging to the three fold-types. (b) Density plot for KR-bias of the last helix of the N- and C-terminal core subdomain belonging to three fold-types. $N_{in}$ and $N_{out}$ core subdomains of the three fold-types are shown as cartoon representations. The last helix shows the change in orientation. $N_{in}$ and $N_{out}$ denote the orientation

of the first helix of the core subdomain. (c) The first four figures show the N- and C-terminal core subdomains of the broken and reentrant transporter. The last helix is shown dark while the other two helices are shown in transparent colour. The final figure shows the structure superimposition between the broken and reentrant core domains.
(PDF)

**S5 Fig. KR bias plots.** (a) KR bias plots showing the orientation of Glt_symporter. (b) KR bias plots showing the orientation of AbrB.
(PDF)

**S6 Fig.** Sequence and topology alignments between Glt_symporter and AbrB families (a) Sequence and topology alignment between Glt_symporter N-terminal repeat and AbrB C-terminal repeat. (b) Sequence and topology alignment between Glt_symporter N-terminal repeat and AbrB N-terminal repeat.
(PDF)

**S7 Fig. Sequence and topology alignment between SBF_1 and Mem_trans families.** (a) Sequence and topology alignment between SBF_1 and Mem_trans N-terminal repeat units. (b) KR-bias plots showing the orientation of the SBF_like family. (c) KR-bias plots showing the orientation of the Mem_trans family.
(PDF)

**S8 Fig. Sequence and topology alignment between full-length Na_H_exchanger_1 and Na_H_Exchanger_2 subfamilies (Same fold-type).**
(PDF)

**S9 Fig. Sequence and topology alignment between pairs of families within the same fold-type containing full-length Glt_symporter and 2HCT.**
(PDF)

**S10 Fig. Sequence and topology alignment between Na_H_Exchanger_1 and KdgT families.** (a) Sequence and topology alignment between pairs of families from different fold-types containing full-length Na_H_Exchanger_1 and KdgT. (b)Structure superposition of families belonging to two families with known structure (PDB id: 4n7w, 4bwz). The gain of helices in one of the broken transporters is highlighted in brown.
(PDF)

**S1 Table. Information about the MSAs for each family.** Meff score calculated for each family MSA, the Pcons-score, the TMscore for the proteins with resolved structure, and the ratio of contacts satisfied in the resulting model. The ratio of the satisfied contact was calculated for a number of contacts equal to the length of the proteins (L) considering the contacts distant more than 5 residues in the sequence.
(PDF)

**S2 Table. Identification of Broken or reentrant type of transporter.** A structural template with the lowest E-value (in bold) is used to identify the broken or reentrant type. Additionally, identification of the type of transporters based on KR bias calculations is chosen based on the highest mean KR bias value (shown in bold). Two E-values are provided for the broken structure template of the Mem_trans family. Mem_trans does not align full length with the structure template. It instead aligns with the repeats in N- C-terminal way.
(PDF)

## Acknowledgments

We thank David Drew for his valuable input to this manuscript. We thank the Swedish National Infrastructure for Computing (https://snic.se) for providing computational resources.

## Author Contributions

**Conceptualization:** Govindarajan Sudha, Claudio Bassot, Arne Elofsson.

**Data curation:** Claudio Bassot.

**Formal analysis:** Govindarajan Sudha, Claudio Bassot.

**Investigation:** Govindarajan Sudha, Claudio Bassot.

**Methodology:** Govindarajan Sudha, Claudio Bassot.

**Software:** Govindarajan Sudha, Claudio Bassot, John Lamb, Nanjiang Shu, Yan Huang.

**Supervision:** Govindarajan Sudha, Arne Elofsson.

**Validation:** Arne Elofsson.

**Visualization:** Govindarajan Sudha, Claudio Bassot, Nanjiang Shu, Arne Elofsson.

**Writing – original draft:** Govindarajan Sudha, Claudio Bassot.

**Writing – review & editing:** Govindarajan Sudha, Claudio Bassot, John Lamb, Nanjiang Shu, Arne Elofsson.

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
