## [Decision Letter · Decision Letter 0]

11 Mar 2021

Dear Dr. Elofsson,

Thank you very much for submitting your manuscript "The evolutionary history of topological variations in the CPA/AT superfamily" for consideration at PLOS Computational Biology.

As with all papers reviewed by the journal, your manuscript was reviewed by members of the editorial board and by several independent reviewers. In light of the reviews (below this email), we would like to invite the resubmission of a significantly-revised version that takes into account the reviewers' comments.

The consensus among the reviewers was that the two manuscripts on the evolution of CPA/AT proteins and the identification of a novel fold type in CPA/AT proteins were each of insufficient contribution on their own. However, if the two manuscripts were combined and the other revisions implemented, they may be of sufficient interest for publication. We of course understand if you wish to publish these manuscripts as two separate articles in a different venue. In addition, I would like to apologize for the length of time it took to complete reviews on this manuscript. I wish it could have been shorter.

We cannot make any decision about publication until we have seen the revised manuscript and your response to the reviewers' comments. Your revised manuscript is also likely to be sent to reviewers for further evaluation.

Sincerely,

Joanna Slusky, Ph.D.

Guest Editor

PLOS Computational Biology

Feilim Mac Gabhann

Editor-in-Chief

PLOS Computational Biology

The consensus among the reviewers was that the two manuscripts on the evolution of CPA/AT proteins and the identification of a novel fold type in CPA/AT proteins were each of insufficient contribution on their own. However, if the two manuscripts were combined and the other revisions implemented, they may be of sufficient interest for publication.

Reviewer's Responses to Questions

**Comments to the Authors:**

Reviewer #1: The current study focuses on the CPA/AT family of transmembrane transporters and proposes a different mode of classification based on a combination of evolutionary, biochemical and structural (topology) features that provides a more in-depth view of the differences between the members of this group of proteins. The study also collects and organizes these results into a searchable database developed by the authors.

The work done on the CAP/AT family is extensive and uses a combination of high throughput computational analysis and some manual verification and corrections required when dealing with large number of sequences and structures.

As a whole, I believe this manuscript is a valuable contribution to the structural biology, classification and evolution of membrane proteins as it paints a renovated picture of what it constitutes to be a member of this family by including new ideas like “fold types” and topology prediction. I was less enthusiastic when thinking about the potential use of this study or the database in the community. I would have liked to see examples on how the community could benefit from the results of this new way of classifying the family and how the proposed models of evolution can be used. If the authors could include a section clarifying or illustrating these possibilities, I believe, it would strengthen this manuscript.

I have a series of comments and questions which I would like to be addressed in future versions of this manuscript:

General Comments

1. The authors should do a better job convincing the reader why yet another classification of transporters is needed. Pointing out discrepancies between the current classifications is not enough since this new classification will bring yet another set of discrepancies.

2. In the beginning of the results section it is explained that two new families were added, an explanation was provided alluding to the need of reliable alignment to other families, however it is not explained why. Conversely, two families were excluded because they are distantly related, how is threshold for distance defined? And why are those families present in the family in the first place?

3. Figure 2, deals with the family DUF819 family. I would suggest to incorporate this into the actual figure to make it clear for the reader when looking at the figure before reading the captions

4. The KR-bias is used in Fig. 2 before it is properly introduced in the manuscript. I suggest you include a brief definition in the figure caption to correct this continuity issue.

5. The MSA-MSA pairwise alignment is not clearly defined. Please include a brief explanation in the fold types section.

6. The manuscript uses the concept of the “positive inside rule” as a ground truth, is this assumption extensive and general? As a non-expert I would doubt this always has to be true, could the authors clarify ?

7. In step 1 of topology annotations, the authors use the reference proteome sequences for Pfam. Typically this set is much smaller than those including all the Uniprot proteins. What is the rationale to use only the reference proteomes and not all available sequences?

8. Please include a section describing possible applications or use cases of this database. For example Pfam can be used as input MSAs for contact prediction, what about this database?

Minor details

1) Abstract. Replace “losses of core helices” with “loss of core helices”

2) Author summary. Line 60, add a period between “superfamily” and “We”

3) Page 5, line 101. Replace “repeat (In ..” With “repeat (in ..”

4)Page 6, line 115 , replace “enter and exit” with “enters and exits”

5) In table one some cells start with a semicolon (;) what is the reason for this?

6) Fig. 3 captions, line 193, replace “is represented” with “are represented”

7) Page 12, line 210, replace “helices themself” with “helices themselves” or remove “themself”

8) Page 14, line 238. I suggest to remove “Anyhow” at the beginning of the paragraph

9) Page 17, line 281. Replace “family have” with “family has”

10) Page 21, line 374. Replace the comma with a period in “selected, In case”

11) Could the authors explain what is the PDBmmCIF70_22_May database?

Reviewer #2: In the submitted manuscript authors present sequence- and structure-based classification of the CPA/AT transporters superfamily. Such a classification is an important step towards understanding the evolution of these proteins, however, experiments being at the base of this work may lack robustness.

- Fig. 2. “The figure shows a reordered topology alignment” – what does reordered mean in this context? What is the meaning of helix numbering (also in Figs. 4, 5, and 6). Is the presence of the reentrant helix (RH) in N or C repeat mutually exclusive? These issues should be addressed in the introduction. Also, figures’ legends would benefit from clarifications. Now it is hard to understand them without referring to the methods.

- It is unclear to me how Fig. 3 was used to define the three groups. First, the title of the figure is “Repeat units are clustered based on conserved C-terminal repeat” (also Pg. 19, Ln. 320) but it shows 30 units which correspond (I guess) to N and C terminal repeats of the 15 families from Table 1 (this is also stated on page 10, lines 182-183). It is not clear when the authors refer to a single (N or C) repeat or the full fusion protein. Second, I don’t understand how the investigation of the dendrogram could have led to the definition of the groups. The purple group actually comprises two sub-groups that share no similarity, the same is for the green and pink groups (I don’t see how Fig 3 could support their existence – page 11, lines 196-197). I suggest representing the all-vs-all similarities in a form of a cluster map (e.g., with CLANS [T Frickey 2004] or a similar tool) and using such a representation for defining groups. Currently, it looks like the manually-defined groups were mapped onto the dendrogram just to state that they were actually obtained from it.

- It is not clear what Fig. S6 actually shows. Please clarify. Pg. 12, Ln. 212-214. Why helix gain/loss would be restricted only to the C-termini scaffold subdomain? Can’t the broken to reentrant transition occur in the N-termini core domain? Pg. 14, Ln. 242-250 – why only one example was given? Are there other examples that would support different scenarios (for example, a transition that does not involve duplication of a single repeat but rather a transformation of a two-repeat fusion protein)?

- In general, the presented evolutionary scenarios (Pg. 12-17) are speculative and are based on anecdotical observations rather than the investigation of all the available data (comparison of the repeating units). In my opinion, the work would benefit from having a figure in the main text (maybe a schematic tree?) that would summarize all the families, including the relative similarities between their repeats, their features (core helix type, orientation, etc.), and list the possible scenarios of duplications/fusions/swaps, etc. The S16 Fig. may be a good starting point for such a figure.

- It is not clear why the four families were excluded from the analyses.

- The authors do not comment on the fact that in some cases the two repeats are not in a single protein chain but are rather encoded separately.

Reviewer #3: The work described by Sudha and colleagues, describes the effort to use a number of sequence and structural bioinformatics tools, to study the evolutionary history of the CPA/AT family.

Clearly this work, is relevant for a very specialized crowd that studies the CPA/AT superfamily and I have a hard time seeing the impact of this work outside of that community. On a more positive note, the authors di make the effort to summarize their findings on a searchable database, that can serve the community that studies these proteins to use the findings of this paper.

Another point that strikes me in the manuscript is related to the proposed evolutionary pathways – as far as my understanding goes these evolutionary pathways are just “putative solutions” and sometimes the language in the manuscript oscillates in between this is a proposed mechanism and this is the only event possible. Would be important to be consistent in this section.

Specific comments

CPA/AT family in the title is generally not a great way to present the work to a broader audience as many scientists will not know what the CPA/AT superfamily is

- line 60 missing a punctuation

-line 63 would or could ?

- in terms of notation I find figure 1 c confusing as clearly we do not have 2 and 3 helices – but rather 4 and 6

**Have all data underlying the figures and results presented in the manuscript been provided?**

Reviewer #1: Yes

Reviewer #2: Yes

Reviewer #3: Yes

PLOS authors have the option to publish the peer review history of their article (what does this mean?). If published, this will include your full peer review and any attached files.

Reviewer #1: No

Reviewer #2: No

Reviewer #3: No
---

## [Decision Letter · Decision Letter 1]

9 Jun 2021

Dear Dr. Elofsson,

Thank you very much for submitting your manuscript "The evolutionary history of topological variations in the CPA/AT transporters" for consideration at PLOS Computational Biology. As with all papers reviewed by the journal, your manuscript was reviewed by members of the editorial board and by several independent reviewers. The reviewers appreciated the attention to an important topic. Based on the reviews, we are likely to accept this manuscript for publication, providing that you modify the manuscript according to the review recommendations.

Thank you especially for your flexibilty with this manuscript.

Sincerely,

Joanna Slusky, Ph.D.

Guest Editor

PLOS Computational Biology

Feilim Mac Gabhann

Editor-in-Chief

PLOS Computational Biology

[LINK]

Reviewer's Responses to Questions

**Comments to the Authors:**

Reviewer #1: The authors have made an extensive revision to the original article(s). They have responded in a satisfactory way my questions and requests for clarification, as well to other reviewers’ comments. I have only a couple of comments that I hope can be incorporated in a final version of this article. These proposed changes do not require an additional revision.

1. The authors responded in a clear way potential use cases and applications of the database (Question 8), but I did not see this information incorporated in the final text. A little bit of this response is incorporated in Data availability but it is not complete. The authors should make sure to incorporate this information as it provides a guideline to general audiences about the importance of this work. A partial incorporation that helps retain style and continuity would be ok with me.

2. Make sure the correct symbol is used for the biological hydrophobicity scale, at the moment the symbol appears as a broken symbol in the PDF.

Reviewer #2: All my comments and suggestions were addressed what is much appreciated. In my opinion, the merged manuscript reads very well (both the results and methods sections). I have just a few minor suggestions:

Pg. 4. HHpred is not a tool for homology modeling but for template identification.

Pg. 4. “Most importantly, we identify a novel fold, named the reentrant-helix-reentrant fold, present in three families and describe their evolutionary history.” – I suggest adding a sentence giving a glimpse on what is new/interesting/unusual in the new fold (lack of symmetry?).

Figure 3b. Node labels are too crowded and thus hard to read. Perhaps the authors could label nodes with numbers and provide a legend?

Pg. 8. “The network in Figure 5 clearly shows the evolutionary relationship” – Figure 3, I guess? Please check the figures numbering.

Pg. 11, point 4. Font type/size seems to be different

Pg. 12. “can evolve independently,, shuffle” – two commas?

Final thought: most of the presented evolutionary scenarios involve an intermediate state corresponding to a non-functional protein (lack of the active site, broken symmetry, etc.). I am wondering about the mechanism that enabled their “resurrection”. Paralogs with a similar function? Formation of inter-family hybrids?

**Have the authors made all data and (if applicable) computational code underlying the findings in their manuscript fully available?**

Reviewer #1: Yes

Reviewer #2: Yes

PLOS authors have the option to publish the peer review history of their article (what does this mean?). If published, this will include your full peer review and any attached files.

Reviewer #1: No

Reviewer #2: No

Figure Files:

Data Requirements:

Reproducibility:

References:

---

## [Editor Report · Decision Letter 2]

14 Jul 2021

Dear Dr. Elofsson,

We are pleased to inform you that your manuscript 'The evolutionary history of topological variations in the CPA/AT transporters' has been provisionally accepted for publication in PLOS Computational Biology.

Best regards,

Joanna Slusky, Ph.D.

Guest Editor

PLOS Computational Biology

Feilim Mac Gabhann

Editor-in-Chief

PLOS Computational Biology

---

## [Editor Report · Acceptance letter]

11 Aug 2021

PCOMPBIOL-D-20-02296R2 

The evolutionary history of topological variations in the CPA/AT transporters

Dear Dr Elofsson,

I am pleased to inform you that your manuscript has been formally accepted for publication in PLOS Computational Biology. Your manuscript is now with our production department and you will be notified of the publication date in due course.

With kind regards,

Andrea Szabo
